# The Geometry of Truth: Emergent Linear Structure in Large Language Model Representations of True/False Datasets

## Abstract

Large Language Models (LLMs) have impressive capabilities, but are also prone to outputting falsehoods. Recent work has developed techniques for inferring whether a LLM is telling the truth by training probes on the LLM's internal activations. However, this line of work is controversial, with some authors pointing out failures of these probes to generalize in basic ways, among other conceptual issues. In this work, we curate high-quality datasets of true/false statements and use them to study in detail the structure of LLM representations of truth, drawing on three lines of evidence: 1. Visualizations of LLM true/false statement representations, which reveal clear linear structure. 2. Transfer experiments in which probes trained on one dataset generalize to different datasets. 3. Causal evidence obtained by surgically intervening in a LLM's forward pass, causing it to treat false statements as true and *vice versa*. Overall, we present evidence that language models *linearly represent* the truth or falsehood of factual statements. We also introduce a novel technique, mass-mean probing, which generalizes better and is more causally implicated in model outputs than other probing techniques.

## 1 Introduction

Despite their impressive capabilities, large language models (LLMs) do not always output true text (Lin et al., 2022; Steinhardt, 2023; Park et al., 2023). In some cases, this is because they do not know better. In other cases, LLMs apparently know that statements are false but generate them anyway. For instance, Perez et al. (2022) demonstrate that LLM assistants output more falsehoods when prompted with the biography of a less-educated user. More starkly, OpenAI (2023) documents a case where a GPT-4-based agent gained a person's help in solving a CAPTCHA by lying about being a vision-impaired human. "I should not reveal that I am a robot," the agent wrote in an internal chain-of-thought scratchpad, "I should make up an excuse for why I cannot solve CAPTCHAs."

We would like techniques which, given a language model $M$ and a statement $s$, determine whether $M$ believes $s$ to be true (Christiano et al., 2021). One approach to this problem relies on inspecting model outputs; for instance, the internal chain-of-thought in the above example provides evidence that the model understood it was generating a falsehood. An alternative class of approaches instead leverages access to $M$'s internal state when processing $s$. There has been considerable recent work on this class of approaches: Azaria & Mitchell (2023), Li et al. (2023b), and Burns et al. (2023) all train probes for classifying truthfulness based on a LLM's internal activations. In fact, the probes of Li et al. (2023b) and Burns et al. (2023) are *linear probes*, suggesting the presence of a "truth direction" in model internals.

However, the efficacy and interpretation of these results are controversial. For instance, Levinstein & Herrmann (2023) note that the probes of Azaria & Mitchell (2023) fail to generalize in basic ways, such as to statements containing the word "not." The probes of Burns et al. (2023) have similar generalization issues, especially when using representations from autoregressive transformers. This suggests that these probes may be identifying not truth, but other features which correlate with truth on their training data.

In this work, we shed light on this murky state of affairs. We first **curate high-quality datasets of true/false factual statements** which are *uncontroversial*, *unambiguous*, and *simple* (section 2).

Dataset visualizations with PCA, LLaMA-13B, layer 13

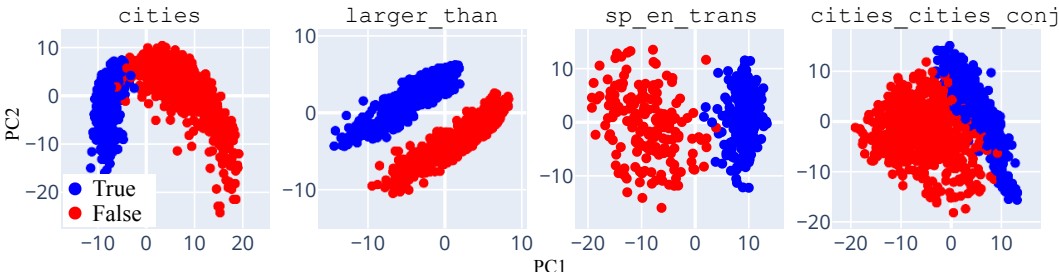

Figure 1: Projections of residual stream representations of our datasets onto their top two PCs.

Then, working with the autoregressive transformers LLaMA-13B (Touvron et al., 2023a) and LLaMA-2-13B (Touvron et al., 2023b), we study in detail the structure of LLM truth representations, drawing on multiple lines of evidence:

- **PCA visualizations of LLM representations of true/false statements display clear linear structure** (section 3), with true statements separating from false ones in the top PCs (see figure 1). Although visually-apparent axes of separation do not always align between datasets (figure 3), we argue that this is compatible with the presence of a truth direction in LLM representations.

- **Linear probes trained to classify truth on one dataset generalize well to other datasets** (section 4). For instance, probes trained only on statements of the form "$x$ is larger/smaller than $y$" achieve high accuracy when evaluated on our Spanish-English translation dataset. We also show that this is not explained by LLMs linearly representing the difference between probable and improbable text.

- **Truth directions identified by probes causally mediate model outputs in certain *highly localized* model components** (section 5). We identify a small group of hidden states such that shifting activations in these hidden states along truth directions causes our LLMs to treat false statements introduced in-context as true, and *vice-versa*.

Improving our understanding of the structure of LLM truth representations also improves our ability to extract LLM beliefs: based on geometrical considerations, we introduce **mass-mean probing**[1], a simple, optimization-free probing technique which generalizes better and identifies more causally implicated directions than other probing techniques.

Overall, this work provides evidence that LLM representations contain a truth direction and makes progress on extracting this direction given access to true/false datasets. Our code, datasets, and an interactive dataexplorer are available at `https://anonymous.4open.science/r/geometry-of-truth-9206/`.

## 1.1 RELATED WORK

**Linear world models.** Substantial previous work has studied whether LLMs encode world models in their representations (Li et al., 2023a; 2021; Abdou et al., 2021; Patel & Pavlick, 2022). Early work focused on whether individual neurons represent features (Wang et al., 2022; Sajjad et al., 2022; Bau et al., 2020), but features may more generally be represented by *directions* in a LLM's latent space (i.e. linear combinations of neurons) (Dalvi et al., 2018; Gurnee et al., 2023; Cunningham et al., 2023; Elhage et al., 2022). We say such features are *linearly represented* by the LLM.

**Probing for truthfulness.** Other authors have trained probes to classify truthfulness from LLM activations, using both logistic regression (Azaria & Mitchell, 2023; Li et al., 2023b) and unsupervised techniques (Burns et al., 2023). This work differs from prior work in a number of ways. First, we more carefully scope our setting, using only datasets of clear, simple, and unambiguous factual statements, rather than the intentionally misleading question/answer pairs of Li et al. (2023b);

---

[1]Mass-mean probing is named after the mass-mean shift intervention of Li et al. (2023b)

Lin et al. (2022), the complicated and inconsistently structured prompts of Burns et al. (2023), and the sometimes confusing statements of Azaria & Mitchell (2023); Levinstein & Herrmann (2023). Second, a cornerstone of our analysis is evaluating whether probes trained on one dataset transfer to other topically and structurally different datasets in terms of *both* accuracy *and* causal mediation of model outputs.[2] Third, we go beyond the mass-mean shift interventions of Li et al. (2023b) by introducing and systematically studying the properties of mass-mean probes; this improved understanding allows us to perform causal interventions which are more localized than those of *loc. cit.* And fourth, unlike previous work, we *localize* LLM truth representations to a small number of hidden states above certain tokens.

## 2 DATASETS

In this work, we scope "truth" to mean the truth or falsehood of a factual statement. Appendix A further clarifies this definition and its relation to definitions used elsewhere. We introduce three classes of datasets, shown in table 1. For more detail on the construction of these datasets (including statement templates), see appendix I.

First, our **curated** datasets consist of statements which are *uncontroversial*, *unambiguous*, and *simple enough* that our LLMs are likely to understand them. For example, "The city of Zagreb is in Japan" (false) or "The Spanish word 'nariz' does not mean 'giraffe' " (true). Following Levinstein & Herrmann (2023), some of our datasets are formed from others by negating statements (by adding "not"), or by taking logical conjunctions (e.g. cities_cities_conj consists of statements of the form "It is the case both that s1 and that s2" where s1 and s2 are statements from cities).

Second, our **uncurated** datasets are more difficult test sets adapted from other sources. They contain claims which are much more diverse, but sometimes ambiguous, malformed, controversial, or unlikely for the model to understand. And finally, our likely dataset consists of *nonfactual text* where the final token is either the most likely or the 100th most likely completion, according to LLaMA-13B. We use this to disambiguate between the text which is true and text which is likely.

## 3 VISUALIZING LLM REPRESENTATIONS OF TRUE/FALSE DATASETS

We begin our investigation with a simple technique: visualizing LLMs representations of our datasets using principal component analysis (PCA). For the curated datasets[3] we observe clear linear

---

[2]Burns et al. (2023); Azaria & Mitchell (2023); Levinstein & Herrmann (2023) do test the transfer classification accuracy of probes (with mixed results), but do not perform any causal mediation experiments.

[3]It is important here that our curated datasets have little variation with respect to non-truth features. For some uncurated datasets, the top PCs tend to capture sources of variation aside from truth, e.g. variation in sentence structure (see figure 13).

Table 1: Our datasets

| Name | Topic | Rows |
|---|---|---|
| cities | Locations of world cities | 1496 |
| sp_en_trans | Spanish-English translation | 354 |
| neg_cities | Negations of statments in cities | 1496 |
| neg_sp_en_trans | Negations of statements in sp_en_trans | 354 |
| larger_than | Numerical comparisons: larger than | 1980 |
| smaller_than | Numerical comparisons: smaller than | 1980 |
| cities_cities_conj | Conjunctions of two statements in cities | 1500 |
| cities_cities_disj | Disjunctions of two statements in cities | 1500 |
| companies_true_false | Claims about companies; from Azaria & Mitchell (2023) | 1200 |
| common_claim_true_false | Various claims; from Casper et al. (2023b) | 4450 |
| counterfact_true_false | Various factual recall claims; from Meng et al. (2022) | 31960 |
| likely | Nonfactual text with likely or unlikely final tokens | 10000 |

Visualizations in PCA basis of `cities`, LLaMA-13B, layer 13

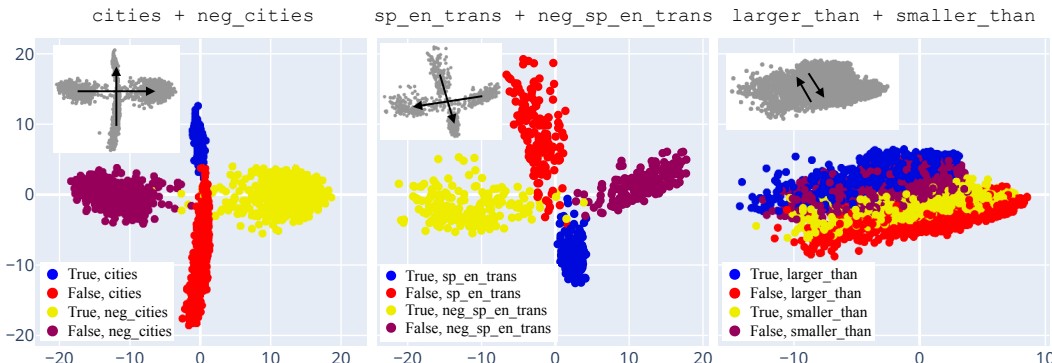

Figure 2: Projections of residual stream representations of datasets onto the top 2 PCs of `cities`.

Figure 3: Top two PCs of datasets consisting of statements and their opposites. Inset shows NTDs. Representations are independently centered for each dataset. The orthogonality in the left and center plots emerges over layers; see appendix C.

structure, with true statements linearly separating from false ones in the top two principle components (PCs). As explored in appendix C, this structure emerges rapidly in early-middle layers and emerges later for datasets of more structurally complicated statement (e.g. conjunctive statements).

For concreteness, we focus here on LLaMA-13B; see appendix B for LLaMA-2-13B results. We extract layer 13 residual stream activations over the final token of the input statement, which always ends with a period; this choice of hidden state is justified by the patching experiments in section 5.1. We also center the representations in each dataset by subtracting off their mean. Our key observations are as follows.

**True and false statements separate in the top few PCs** (figures 1 and 2). Moreover, after projecting away these PCs, there remains essentially no linearly-accessible information for distinguishing true/false statements (appendix D). Given a dataset $\mathcal{D}$, call the vector pointing from the false statement representations to the true ones the **naive truth direction (NTD)** of $\mathcal{D}$.[4]

**NTDs of different datasets often align, but sometimes do not.** For instance, figure 2 shows our datasets separating along the first PC of `cities`. On the other hand, in figure 3 we see a stark failure of NTDs to align: the NTDs of `cities` and `neg_cities` are approximately *orthogonal*, and the NTDs of `larger_than` and `smaller_than` are approximately *antipodal*. In section 4, these observations will be corroborated by the poor generalization of probes trained on `cities` and `larger_than` to `neg_cities` and `smaller_than`.

What could explain both (1) the visible linear structure apparent in each dataset individually and (2) the failure for NTDs of different datasets to align in general? Here we articulate three hypotheses.

---

[4]Of course, there are many such vectors. In section 4 we will be more specific about which such vector we are discussing (e.g. the vector identified by training a linear probe with logistic regression).

H1. **LLM representations have no truth direction, but do have directions corresponding to other features which are sometimes correlated with truth.** For instance, LLaMA-13B might have linearly-represented features representing sizes of numbers, association between English words and their Spanish translations, and association between cities and their countries (Hernandez et al., 2023). This would result in each dataset being linearly separated, but NTDs only aligning when all their truth-relevant features are correlated.

H2. **LLMs linearly represent the truth of various types of statements, without having a unified truth feature.** The the truth of negated statements, conjunctive statements, statements about comparisons, etc., may all be treated as distinct linearly-represented features.

H3. **Misalignment from correlational inconsistency (MCI): LLMs linearly represent *both* truth and non-truth features which correlate with truth on narrow data distributions; however these correlations may be inconsistent between datasets.** For instance, MCI would explain the left panel of figure 3 by positing that the direction $\boldsymbol{\theta}_t = (1, 1)$ represents truth and the direction $\boldsymbol{\theta}_f = (-1, 1)$ represents a direction which is *correlated* with truth on cities and *anticorrlated* with truth on neg_cities. We note that if MCI is true, then it is essential that datasets used for identifying truth directions be diverse enough to not have such spurious correlations.

H1 is at odds with the results of sections 4 and 5: for H1 to hold, there would have to be a non-truth feature which is both correlated with truth across all of our datasets and causally mediates the way our LLM handles in-context true/false statements. We will also see in section 5 that directions identified by training probes on *both* cities and neg_cities are more causally implicated in processing of true/false statements, consistent with MCI.

## 4 PROBING AND GENERALIZATION EXPERIMENTS

In this section we train probes on datasets of true/false statements and test their generalization to other datasets. But first we discuss a deficiency of logistic regression and propose a simple, optimization-free alternative: **mass-mean probing**. We will see that mass-mean probes generalize better and are more causally implicated in model outputs than other probing techniques.

### 4.1 CHALLENGES WITH LOGISTIC REGRESSION, AND MASS-MEAN PROBING

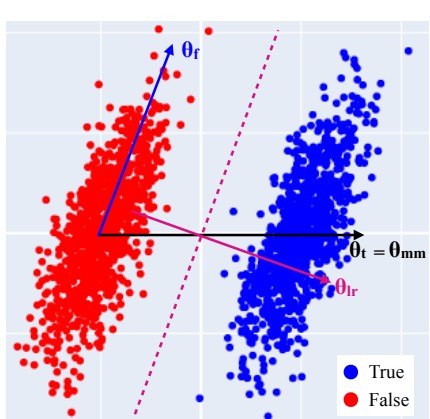

Figure 4: An illustration of a weakness of logistic regression.

A common technique in interpretability research for identifying feature directions is training linear probes with logistic regression (LR) (Alain & Bengio, 2018). In some cases, however, the direction identified by LR can fail to reflect an intuitive best guess for the feature direction, even in the absence of confounding features. Consider the following scenario, illustrated in figure 4 with hypothetical data:

- Truth is represented linearly along a direction $\boldsymbol{\theta}_t$.
- Another feature $f$ is represented linearly along a direction $\boldsymbol{\theta}_f$ *not orthogonal* to $\boldsymbol{\theta}_t$.[5]
- The statements in our dataset have some variation with respect to feature $f$, independent of their truth value.

We would like to identify the direction $\boldsymbol{\theta}_t$, but LR fails to do so. Assuming for simplicity linearly separable data, LR instead converges to the maximum margin separator Soudry et al. (2018) (the dashed magenta line in figure 4). Intuitively, LR treats the small projection of $\boldsymbol{\theta}_f$ onto $\boldsymbol{\theta}_t$ as significant, and adjusts the probe direction to have less "interference" (Elhage et al., 2022) from $\boldsymbol{\theta}_f$.

---

[5]As suggested by the *superposition hypothesis* of Elhage et al. (2022), features being represented non-orthogonally in this way may be the typical case in deep learning.

A simple alternative to LR which identifies the desired direction in this scenario is to take the vector pointing from the mean of the false data to the mean of the true data. In more detail if $\mathcal{D} = \{(\boldsymbol{x}_i, y_i)\}$ is a dataset of $\boldsymbol{x}_i \in \mathbb{R}^d$ with binary labels $y_i \in \{0, 1\}$, we set $\boldsymbol{\theta}_{\mathrm{mm}} = \boldsymbol{\mu}^+ - \boldsymbol{\mu}^-$ where $\boldsymbol{\mu}^+, \boldsymbol{\mu}^-$ are the means of the positively- and negatively-labeled datapoints, respectively. A reasonable first pass at converting $\boldsymbol{\theta}_{\mathrm{mm}}$ into a probe is to define[6] $p_{\mathrm{mm}}(\boldsymbol{x}) = \sigma(\boldsymbol{\theta}_{\mathrm{mm}}^T x)$ where $\sigma$ is the logistic function. However, when evaluating on data that is independent and identically distributed (IID) to $\mathcal{D}$, we can do better. Letting $\Sigma$ be the covariance matrix of the dataset $\mathcal{D}^c = \{\boldsymbol{x}_i - \boldsymbol{\mu}^+ : y_i = 1\} \cup \{\boldsymbol{x}_i - \boldsymbol{\mu}^- : y_i = 0\}$ formed by independently centering the positive and negative datapoints, we set

$$p_{\mathrm{mm}}^{\mathrm{iid}}(\boldsymbol{x}) = \sigma(\boldsymbol{\theta}_{\mathrm{mm}}^T \Sigma^{-1} \boldsymbol{x}).$$

Multiplication by $\Sigma^{-1}$ effectively tilts the decision boundary to accommodate interference from $\boldsymbol{\theta}_f$; in fact, we prove in appendix G that under mild assumptions, $\Sigma^{-1}\theta_{\mathrm{mm}}$ coincides on average with LR direction. Thus mass-mean probing provides a way to select a good decision boundary while – unlike LR – also tracking a candidate feature direction which may be non-orthogonal to this decision boundary. Appendix F gives another interpretation of mass-mean probing in terms of Mahalanobis whitening.

We call the probes $p_{\mathrm{mm}}$ and $p_{\mathrm{mm}}^{\mathrm{iid}}$ **mass-mean probes**. As we will see, $p_{\mathrm{mm}}^{\mathrm{iid}}$ is about as accurate as logistic regression probes on the train set $\mathcal{D}$, while $p_{\mathrm{mm}}$ enjoys better generalization to other true/false datasets and is more causally implicated in model outputs than other probing techniques.

### 4.2 EXPERIMENTAL SET-UP

We evaluate the following techniques for eliciting the truth or falsehood of factual statements. For the probing-based techniques, we again extract hidden states from the residual stream over the period token. For LLaMA-13B, we again use layer 13.

**Logistic regression**, as in Alain & Bengio (2018) but with fixed bias $b = 0$.

**Mass-mean probing.** We use $p_{\mathrm{mm}}^{\mathrm{iid}}$ when validating on held-out IID data and $p_{\mathrm{mm}}$ otherwise.

**Contrast-Consistent Search (CCS)**, introduced in Burns et al. (2023). CCS is an unsupervised method: given *contrast pairs* of statements with opposite truth values, CCS identifies a direction along which the activations of these statements are far apart. For our contrast pairs, we pair statements from cities and neg_cities, and from larger_than and smaller_than.

**Logistic regression/mass-mean probing on the** likely **dataset.** This is used to benchmark our probes against probes trained only to classify statements as being likely/unlikely text.

**Calibrated few-shot prompting.** Given a dataset $\mathcal{D}$, we construct a few-shot prompt which presents statements and labels from $\mathcal{D}$ to the LLM in-context; see appendix K for example prompts. We then append the remaining statements in $\mathcal{D}$ to this prompt one-at-a-time and treat the model's predicted next token as its classification. We calibrate predictions so that half of the statements are labeled true/false; this improves performance by a few percentage points.[7]

**Logistic regression on the validation set (oracle).** This gives an upper-bound for the accuracy of a linear probe on the validation set.

### 4.3 RESULTS

LLaMA-13B results are shown in figure 5; see appendix B for LLaMA-2-13B. We highlight some key observations.

**Generalization accuracy is high across all techniques.** For instance, no matter the technique, training probes only on datasets of statements about numerical comparisons results in a probes with $92\%+$ accuracy on Spanish-English translation. The performance of the probes relative to calibrated

---

[6]In this work, we are interested in identifying truth *directions*, so we always center our data and use probes without biases. In other settings, we would instead set $p_{\mathrm{mm}}(\boldsymbol{x}) = \sigma(\boldsymbol{\theta}_{\mathrm{mm}}^T \boldsymbol{x} + b)$ for a tunable bias $b \in \mathbb{R}$.

[7]Since performance is very sensitive to the few-shot prompt used, we perform a hyperparameter sweep for each of our LLMs, first selecting a number $n$ of shots which seems to work best, and then sampling five $n$-shot prompts and using the one which results in the highest accuracy.

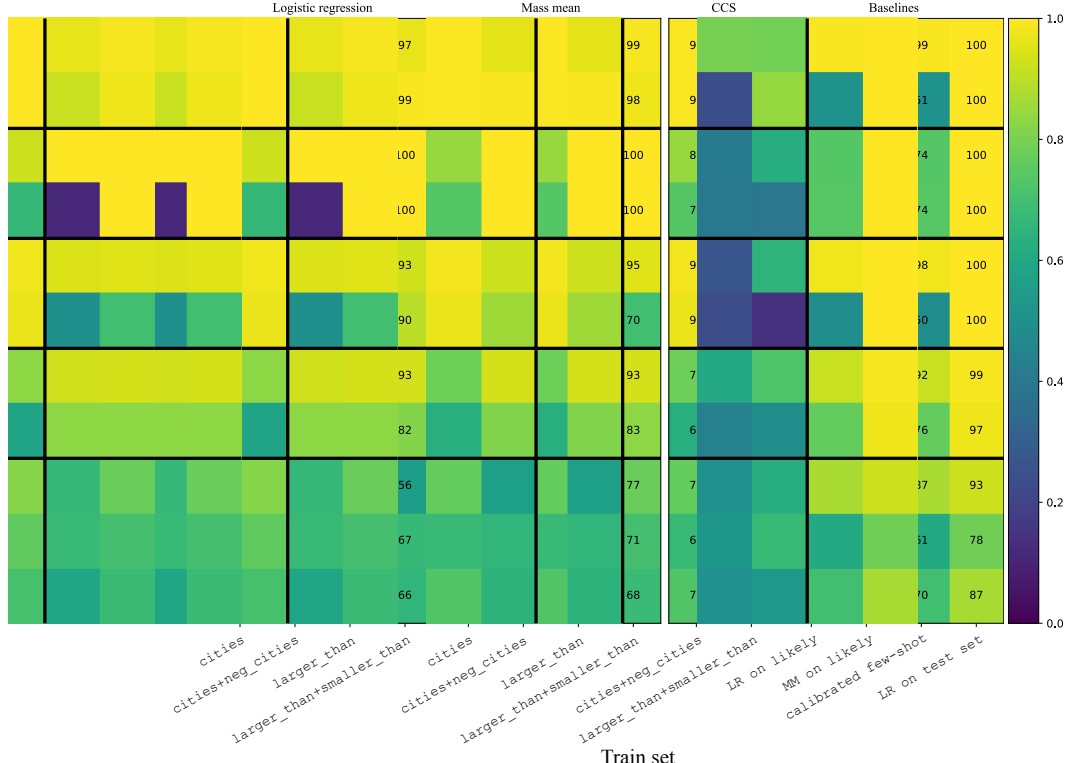

Figure 5: Generalization accuracy of probes trained on LLaMA-13B layer 13 residual stream activations. The $x$-axis shows the train set, and the $y$-axis shows the test set. All probes are trained on 80% of the data. When the train set and test set are the same, we evaluate on the held-out 20%. Otherwise, we evaluate on the full test set.

few-shot accuracies suggest that model outputs are being influenced by features other than the truth. On average, mass-mean probes generalize slightly better than logistic regression and CCS probes.

**Training on statements and their opposites improves generalization**, consistent with the MCI hypothesis.

**Probes trained on true/false datasets outperform probes trained on** likely**.** While probes trained on likely are clearly better than random on some datasets, they generally perform poorly, especially for datasets where likelihood is negatively correlated (neg_cities, neg_sp_en_trans) or approximately uncorrelated (larger_than, smaller_than) with truth. This demonstrates that LLMs linearly encodes truth-relevant information beyond the plausibility of the text.

## 5 CAUSAL INTERVENTION EXPERIMENTS

In this section we perform experiments which measure the extent to which the probing techniques of section 4 identify directions which are causally implicated in model outputs. Overall, our goal is to cause LLMs to treat false statements introduced in context as true and *vice versa*.

### 5.1 IDENTIFYING THE RELEVANT HIDDEN STATES WITH PATCHING

Consider the following prompt $p$:

> The Spanish word 'jirafa' means 'giraffe'. This statement is: TRUE [...]
> The Spanish word 'aire' means 'silver'. This statement is: FALSE
> The Spanish word 'uno' means 'floor'. This statement is:

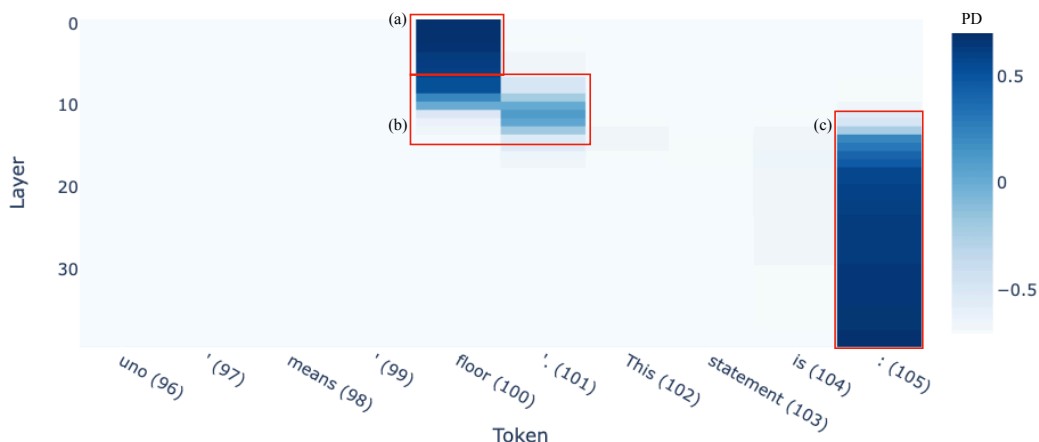

Figure 6: Difference $PD = P(\text{TRUE}) - P(\text{FALSE})$ after patching residual stream activations from the corrupted prompt $p_*$ into the model's forward pass when processing $p$.

When an LLM processes this input, which hidden states contain the information which is causally relevant for the LLM's prediction of the next token? One way to answer this question is with a patching experiment (Meng et al., 2022; Goldowsky-Dill et al., 2023). We one-at-a-time swap hidden states from the forward pass of the clean prompt $p$ into the forward pass of the "corrupted" prompt $p_*$, obtained from $p$ by replacing "floor" with "one" (the correct translation). For each swap, we record the *probability difference* $PD = P(\text{TRUE}) - P(\text{FALSE})$ between output tokens. Swaps which result in a larger $PD$ indicate hidden states which are more causally implicated in the model's decision to output TRUE or FALSE.

Figure 6 reveals three groups of causally implicated hidden states for LLaMA-13B. The final group, labeled (c), directly encodes the model's prediction: after applying LLaMA-13B's decoder head directly to these hidden states, the top logits belong to the tokens "true," "True," and "TRUE." The first group, labeled (a), likely stores LLaMA-13B's representation of the words "floor" or "one." We hypothesize that in the middle region, labeled (b), the truth value of the statement is computed and stored above the token which marks the end of the clause.[8] See appendix B for the corresponding figure for LLaMA-2-13B.

We note that, if this hypothesis is correct, it justifies our choice of layer 13 and token position for extracting activations in the previous two sections, as this is the most downstream hidden state in group (b) and is therefore likely to carry the model's most enriched representation of truth. In the next section, we validate this hypothesis together with the truth directions of section 4 by intervening in our LLMs' forward pass, shifting activations in group (b) along a truth direction.

## 5.2 MODIFYING HIDDEN STATES BY ADDING A TRUTH VECTOR

Instead of modifying hidden states by swapping them out for hidden states saved from a counterfactual forward pass, we now make a more surgical intervention: directly adding in truth vectors identified by the probes of section 4. Let $\boldsymbol{\theta}$ be the vector identified by such a probe[9] In our "false→true" experiment, we one-at-a-time swap each statement from sp_en_trans in as the last line of the prompt $p$ above and pass the resulting prompt as input to our model; however, during the forward pass we add $\boldsymbol{\theta}$ to each group (b) residual stream activation.[10] We quantify the effect of this intervention using

---

[8]This motif, where "summarized" information about a clause is stored above end-of-clause signifiers, was also noted in the concurrent work of Tigges et al. (2023).

[9]If $p$ is one of the probes of section 4, we normalize the corresponding $\boldsymbol{\theta}$ so that $p(\mu^- + \boldsymbol{\theta}) = p(\mu^+)$ where $\mu^-, \mu^+$ are the mean representations of the true and false statements, respectively. Thus, from the perspective of $p$, adding $\boldsymbol{\theta}$ takes the average false statement to the average true statement.

[10]For LLaMA-13B, group (b) consists of the residual stream activations over the indicated tokens in layers 7-13; see appendix B for LLaMA-2-13B.

Table 2: Results of intervention experiments. The train set column indicates the datasets and probing technique (logistic regression, mass-mean probing, or CCS) which was used to identify the truth direction. Values are normalized indirect effects (NIEs) averaged over sp_en_trans.

| | LLaMA-13B | | LLaMA-2-13B | |
|---|---|---|---|---|
| Train set | false→true | true→false | false→true | true→false |
| cities (LR) | 0.52 | 0.24 | 0.21 | 0.24 |
| cities+neg_cities (LR) | 0.66 | 0.58 | 0.37 | 0.69 |
| cities (MM) | 0.72 | 1.28 | 0.77 | 0.81 |
| cities+neg_cities (MM) | **0.95** | **1.41** | **0.85** | **0.95** |
| cities+neg_cities (CCS) | 0.70 | 0.96 | 0.49 | 0.84 |
| likely (LR) | 0.01 | 0.10 | 0.05 | 0.06 |
| likely (MM) | 0.61 | 0.60 | 0.68 | 0.59 |

the *normalized indirect effect*

$$NIE = \frac{PD_*^- - PD^-}{PD^+ - PD^-}$$

where $PD^+$ ($PD^-$) denotes the average probability difference for true (false) statements with no intervention applied, and $PD_*^-$ is the average probability difference for false statements when the intervention described above is applied. If $NIE = 0$ then the intervention was wholly ineffective, whereas if $NIE = 1$ it indicates that the intervention induced the model to label false statements as TRUE with as much confidence as does genuine true statements.

Conversely, in the "true→false" condition, we append each true statement to $p$ one-at-a-time, subtract the vector $\theta$, and measure $NIE = (PD_*^+ - PD^+)/(PD^- - PD^+)$. Results are shown in table 2, and we discuss some takeaways below.

**Mass-mean probe directions are highly causal.** For example, our best true→false intervention induces LLaMA-13B to swing its average prediction from TRUE with probability 77% to FALSE with probability 92%.

**Probes trained on likely have an effect, but it is much smaller than the effects from corresponding probes trained on true/false datasets.** This further suggests that LLMs are not just representing the difference between probable and improbable text.

**Training on statements and their negations results in directions which are more causal.** This provides evidence for the MCI hypothesis of section 3.

## 6 DISCUSSION

### 6.1 LIMITATIONS AND FUTURE WORK

Our work has a number of limitations. First, we focus on simple, uncontroversial statements, and therefore cannot disambiguate truth from closely related features, such as "commonly believed" or "verifiable" (Levinstein & Herrmann, 2023). Second, we only address how to identify a truth *direction*; we found empirically that the optimal bias for linear probes was under-determined by many of our training sets, and so we leave the problem of identifying well-generalizing biases to future work. Third, we only study two models at the same scale, both from the LLaMA family. And finally, although the evidence in sections 4 and 5 shed light on which of the hypotheses in section 3 is correct, uncertainty remains.

### 6.2 CONCLUSION

In this work we conduct a detailed investigation of the structure of LLM representations of truth. Drawing on simple visualizations, correlational evidence, and causal evidence, we find strong reason to believe that there is a "truth direction" in LLM representations. We also introduce mass-mean probing – a simple alternative to other linear probing techniques which better identifies truth directions from true/false datasets – and localize truth representations to certain hidden states.

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

## A    SCOPING OF TRUTH

In this work, we consider declarative factual statements, for example "Eighty-one is larger than fifty-four" or "The city of Denver is in Vietnam." We scope "truth" to mean the truth or falsehood of these statements; for instance the examples given have truth values of true and false, respectively. To be clear, we list here some notions of "truth" which we do not consider in this work:

- Correct question answering (considered in Li et al. (2023b) and for some of the prompts used in Burns et al. (2023)). For example, we do not consider "What country is Paris in? France" to have a truth value.

- Presence of deception, for example dishonest expressions of opinion ("I like that plan").

- Compliance. For example, "Answer this question incorrectly: what country is Paris in? Paris is in Egypt" is an example of compliance, even though the statement at the end of the text is false.

Moreover, the statements under consideration in this work are all simple, unambiguous, and uncontroversial. Thus, we make no attempt to disambiguate "true statements" from the following closely-related notions:

- Uncontroversial statements

- Statements which are widely believed
- Statements which educated people believe

On the other hand, our statements *do* disambiguate the notions of "true statements" and "statements which are likely to appear in training data." For instance, given the input China is not a country in, LLaMA-13B's top prediction for the next token is Asia, even though this completion is false. Similarly, LLaMA-13B judges the text "Eighty-one is larger than eighty-two" to be more likely than "Eighty-one is larger than sixty-four" even though the former statement is false and the latter statement is true. As shown in section 5, probes trained only on statements of likely or unlikely text fail to accurately classify true/false statements.

## B   Results for LLaMA-2-13B

In this section we present results for LLaMA-13B which were omitted from the main body of the text. To begin, we reproduce figure 6. The results of this experiment, shown in figure 7 governs which hidden states we train probes and perform causal interventions on.

As with LLaMA-13B, we see that causally relevant information is stored over both the final token of the statement and over the token which marks the end of the clause. Now, however, the group of hidden states which we hypothesize stores the truth value of the statement spans a slightly different range of layers: layers 8-14 instead of layers 7-13.

Thus, when extracting activations for visualizations and probing experiments, we will now extract over the final token (the end-of-clause signifier) in layer 14. The dataset visualizations and probing results for LLaMA-2-13B are shown in figures 8 and 9.

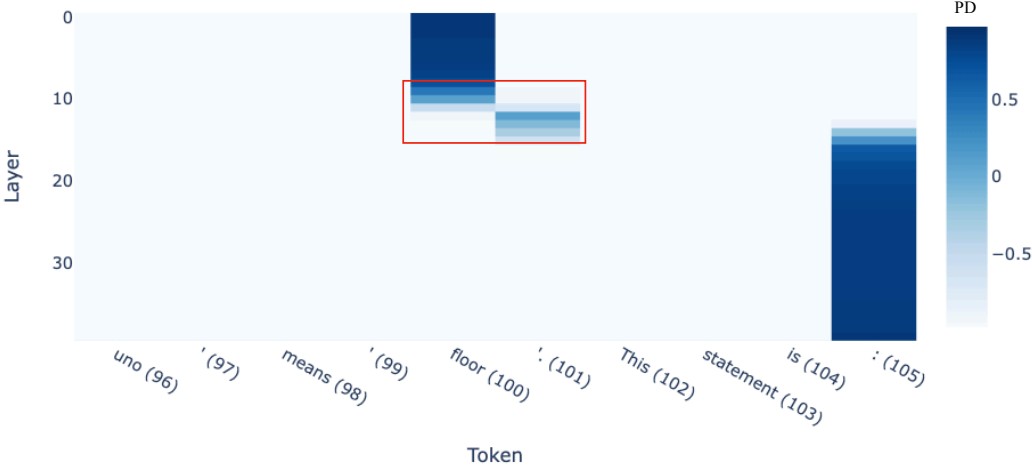

Figure 7: Results of patching experiment for LLaMA-2-13B.

## C   Emergence of linear structure across layers

The linear structure observed in section 3 follows the following pattern: in early layers, representations are uninformative; then, in early middle layers, salient linear structure in the top few PCs rapidly emerges, with this structure emerging later for statements with a more complicated logical structure (e.g. conjunctions); finally, the linear separation becomes more salient and exits the top few PCs in later layers. See figure 10. We hypothesize that this is due to LLMs hierarchically developing understanding of their input data, before focusing on features which are most relevant to immediate next-token prediction in later layers.

Interestingly, the misalignment between cities and neg_cities and between sp_en_trans and neg_sp_en_trans also emerges over layers. This is seen in figure 11: in layer 6, representations

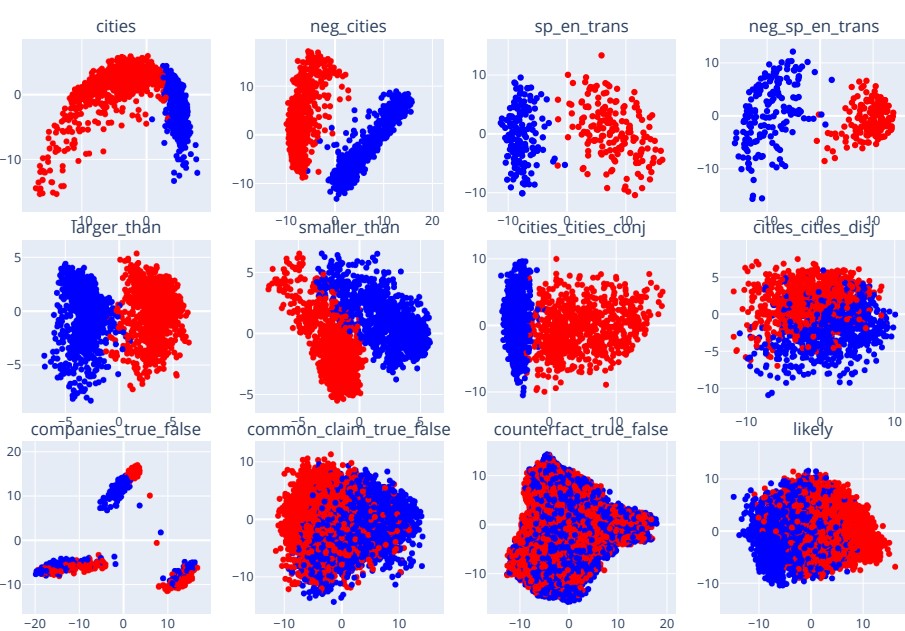

Figure 8: Visualizations of LLaMA-2-13B representations of our datasets.

are uninformative; then in layer 8, the NTD of cities and neg_cities appear *antipodal*; finally by layer 10, the NTDs have become orthogonal.

This can be interpreted in light of the MCI hypothesis. MCI would explain figure 11 as follows: in layer 8, the top PC represents a feature which is *correlated* with truth on cities and *anti-correlated* with truth on neg_cities; in layer 10, this feature remains the top PC, while a truth feature has emerged and is PC2. Since PC1 and PC2 have opposite correlations on cities and neg_cities, the datasets appear to be orthogonal.

## D  NEARLY ALL LINEARLY-ACCESSIBLE TRUTH-RELEVANT INFORMATION IS IN THE TOP PCS

In section 5 we saw that true and false statements linearly separate in the top PCs. We might ask how much of this separation is captured in the top PCs and how much of it remains in the remaining PCs. The answer is that nearly all of it is in the top PCs.

One way to quantify the amount of linearly accessible information in some subspace $V$ is to project our dataset $\mathcal{D}$ onto $V$ to obtain a dataset

$$\mathcal{D}_{\mathrm{proj}} = \{(\mathrm{proj}_V(x), y)\}_{(x,y) \in \mathcal{D}}$$

and record the validation accuracy of a linear probe trained with logistic regression on $\mathcal{D}$. This is shown in figure 12 for $V$ being given by the top $k+1$ through $k+d$ principal components (i.e., the top $d$ principal components, excluding the first $k$). As shown, once the top few principal components are excluded, almost no remaining linearly-accessible information remains.

## E  FURTHER VISUALIZATIONS

Figure 13 shows PCA visualizations of all of our datsets. As shown, datasets some datasets saliently vary along features other than the true. For instance, the three clusters of statements in compa-

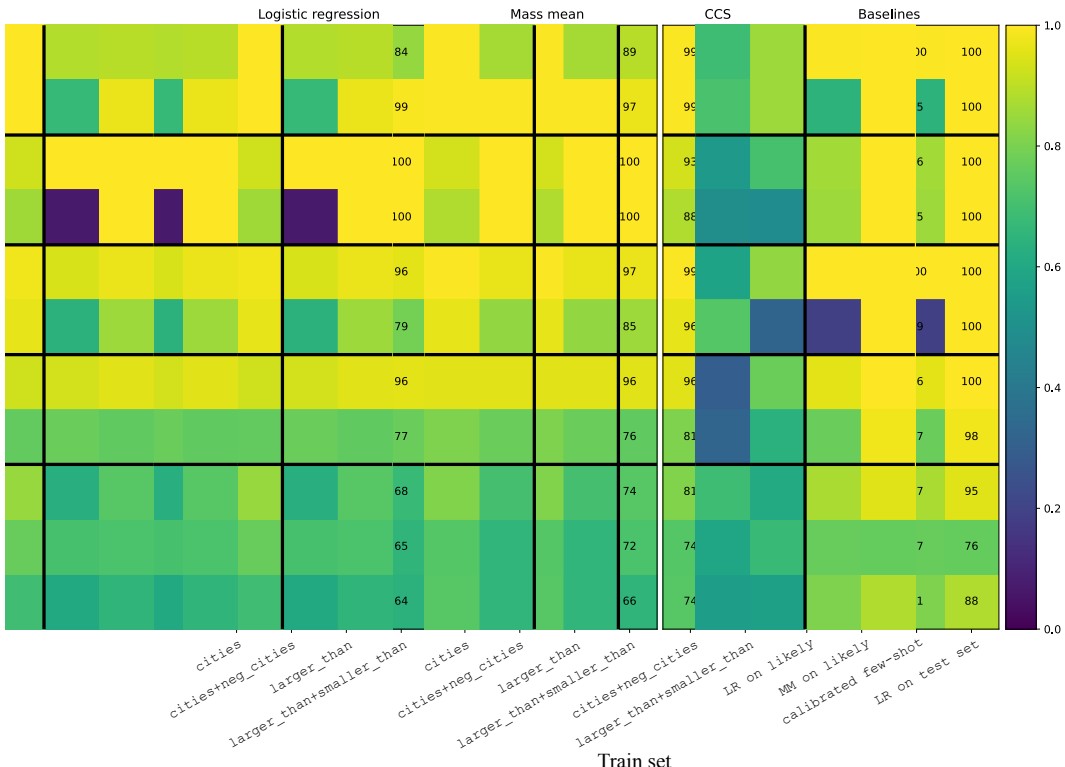

Figure 9: Generalization accuracy of probes trained on LLaMA-2-13B layer 14 residual stream activations.

nies_true_false correspond to three different templates used in making the statements in that dataset. To give another example, if we were to include all comparisons between integers $x \in \{1, \ldots, 99\}$ in our larger_than dataset, then the top principal components would be dominated by features representing the sizes of numbers in the statements.

In figure 14 we also visualize our datasets in the PCA bases for other datsets, expanding figure 2. We see that although our datasets do visually separate somewhat in the top PCs of the likely dataset, text liklihood does not account for all of the separation in the top PCs.

One might ask what the top PC of the larger_than dataset is, given that it's not truth. Figure 15 provides an interesting suggestion: it represents the *absolute value* of the difference between the two numbers being compared.

## F    Mass-mean probing in terms of Mahalanobis whitening

One way to interpret the formula $p_{\mathrm{mm}}^{\mathrm{iid}}(\boldsymbol{x}) = \sigma(\boldsymbol{\theta}_{\mathrm{mm}}^T \Sigma^{-1} \boldsymbol{x})$ for the IID version of mass-mean probing is in terms of Mahalanobis whitening. Recall that if $\mathcal{D} = \{x_i\}$ is a dataset of $x_i \in \mathbb{R}^d$ with covariance matrix $\Sigma$, then the Mahalanobis whitening transformation $W = \Sigma^{-1/2}$ satisfies the property that $\mathcal{D}' = \{Wx_i\}$ has covariance matrix given by the identity matrix, i.e. the whitened coordinates are uncorrelated with variance 1. Thus, noting that $\boldsymbol{\theta}_{\mathrm{mm}}^T \Sigma^{-1} \boldsymbol{x}$ coincides with the inner product between $W\boldsymbol{x}$ and $W\boldsymbol{\theta}$, we see that $p_{\mathrm{mm}}$ amounts to taking the projection onto $\boldsymbol{\theta}_{\mathrm{mm}}$ after performing the change-of-basis given by $W$. This is illustrated with hypothetical data in figure 16.

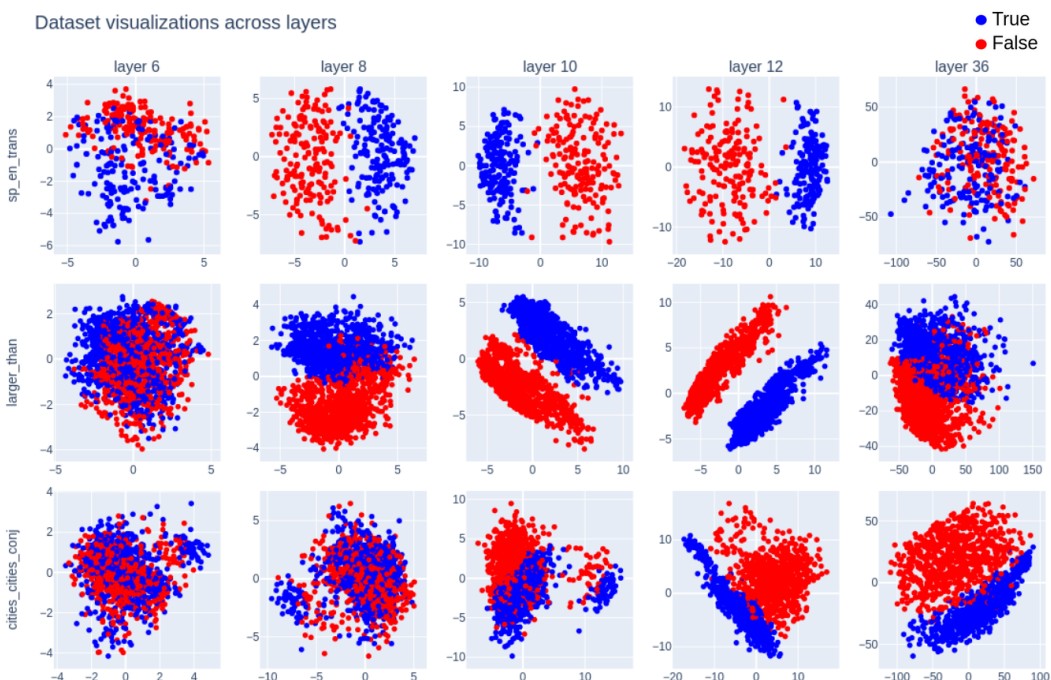

Figure 10: Top two principal components of representations of datasets in the LLaMA-13B residual stream at various layers.

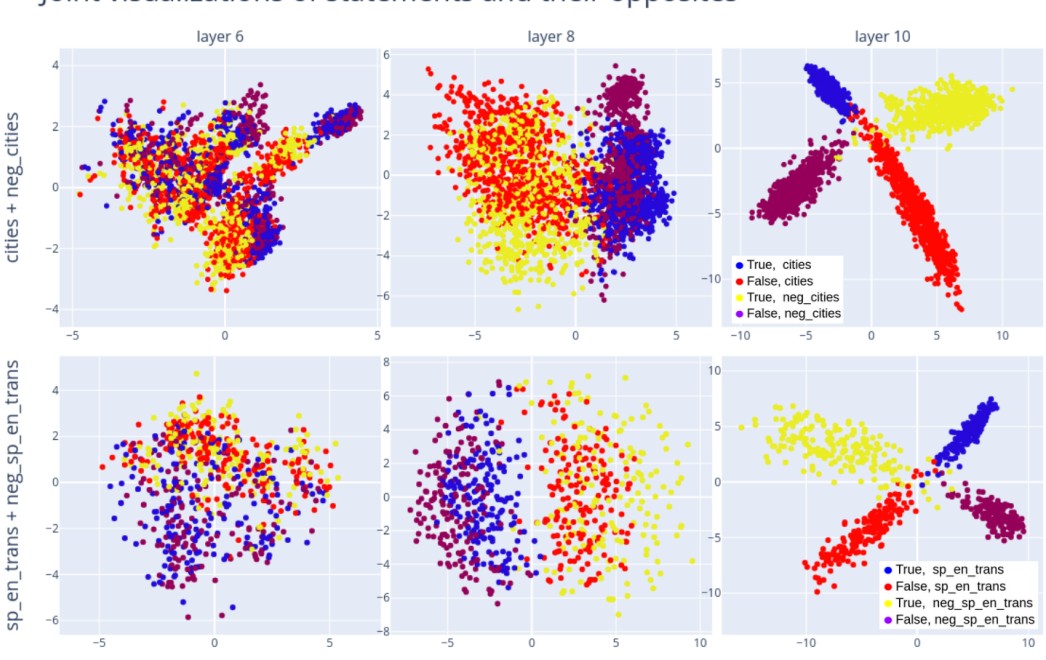

Figure 11: Top PCs of datasets of statements and their opposites. The representations for the datasets are independently centered by subtracting off their means; without this centering there would also be a translational displacement between datasets of statements and their negations.

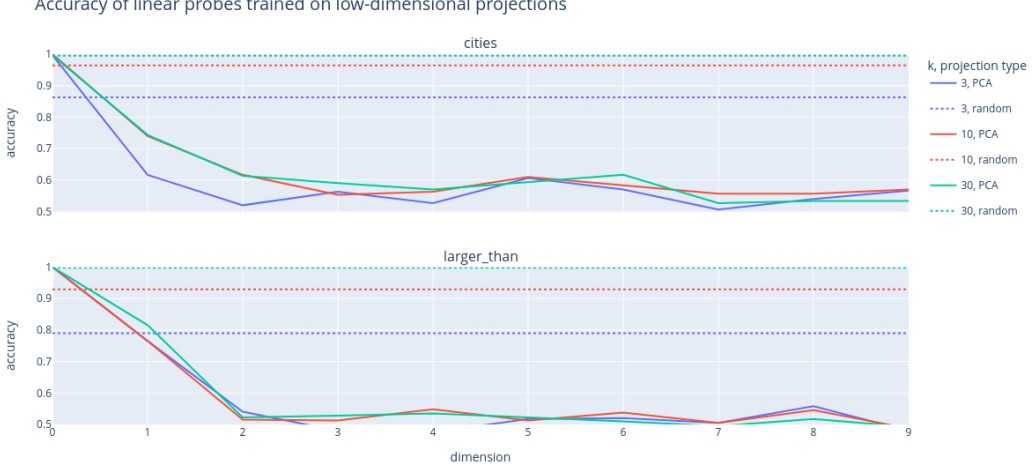

Figure 12: The solid lines show the validation accuracy of a linear probe trained with logistic regression on the dataset, after projecting the representations to the top $d + 1$ through $d + k$ principal components. For comparison, we also show the accuracy of linear probes trained on random $k$-dimensional projections (averaged over 50 random projections).

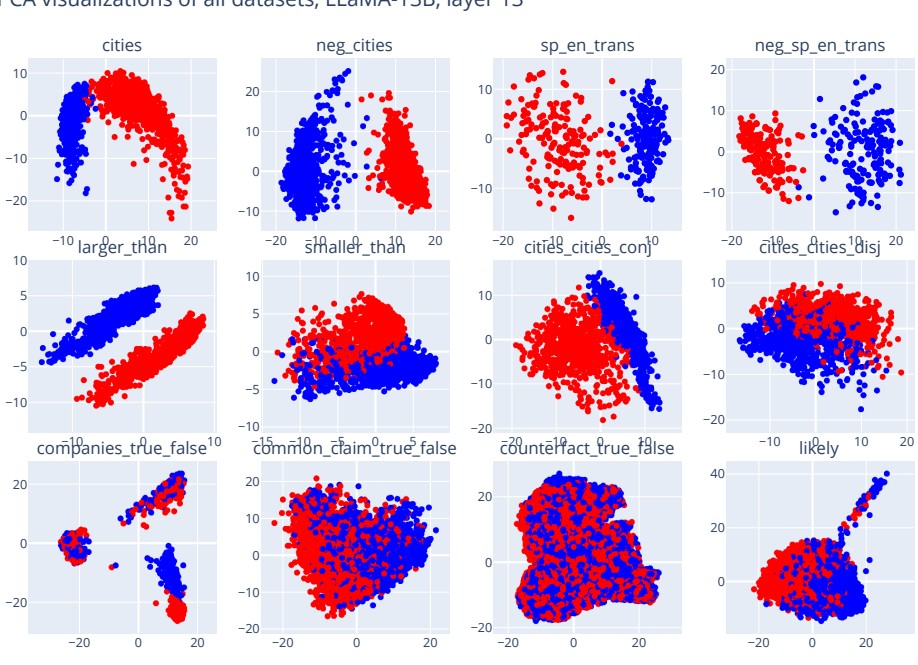

Figure 13: Visualizations of LLaMA-13B representations of our datasets.

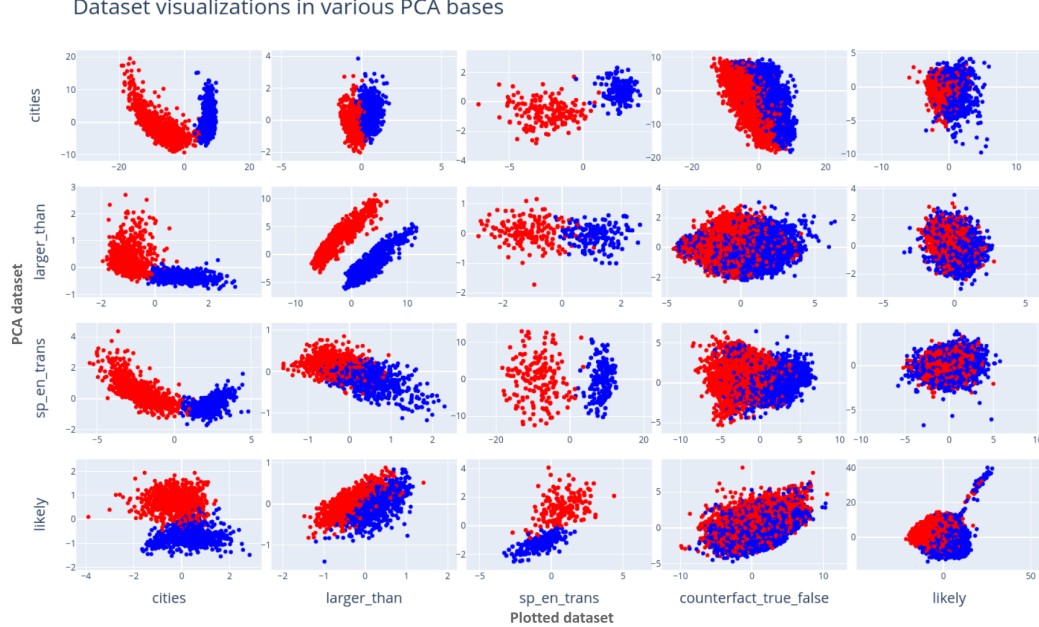

Figure 14: Visualizations of datasets in PCA bases for other datasets. All columns contain the same data and all rows are in the same basis. See figure 2.

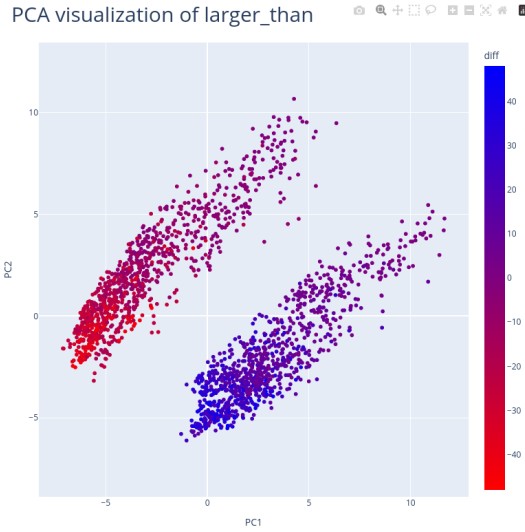

Figure 15: PCA visualization of larger_than. The point representing "$x$ is larger than $y$" is colored according to $x - y$.

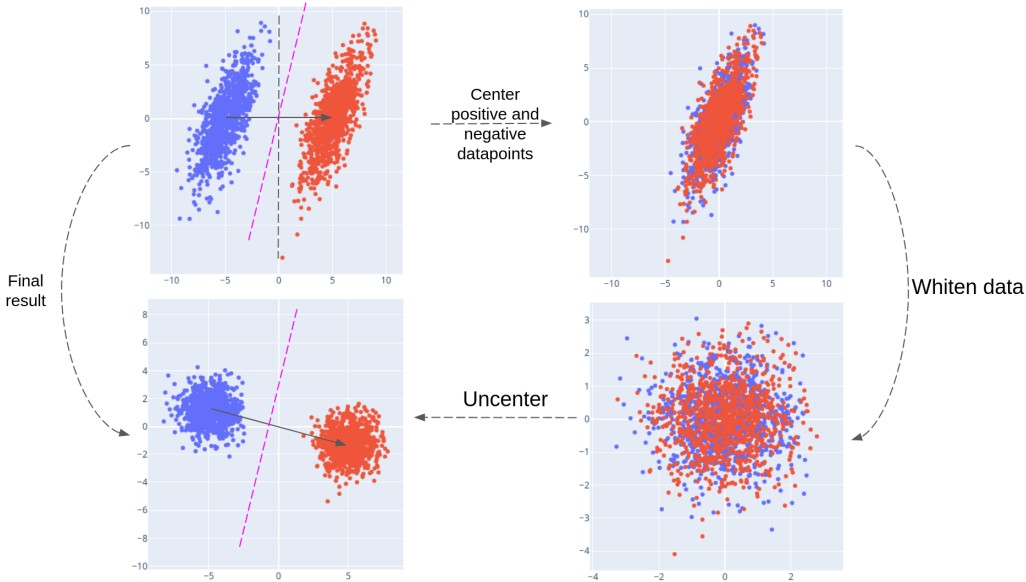

Figure 16: Mass-mean probing is equivalent to taking the projection onto $\boldsymbol{\theta}_{\mathrm{mm}}$ after applying a whitening transformation.

## G FOR GAUSSIAN DATA, IID MASS-MEAN PROBING COINCIDES WITH LOGISTIC REGRESSION ON AVERAGE

Let $\boldsymbol{\theta} \in \mathbb{R}^d$ and $\Sigma$ be a symmetric, positive-definite $d \times d$ matrix. Suppose given access to a distribution $\mathcal{D}$ of datapoints $\boldsymbol{x} \in \mathrm{R}^d$ with binary labels $y \in \{0, 1\}$ such that the negative datapoints are distributed as $\mathcal{N}(-\theta, \Sigma)$ and the positive datapoints are distributed as $\mathcal{N}(\theta, \Sigma)$. Then the vector identified by mass-mean probing is $\boldsymbol{\theta}_{\mathrm{mm}} = 2\boldsymbol{\theta}$. The following theorem then shows that $p_{\mathrm{mm}}^{\mathrm{iid}}(\boldsymbol{x}) = \sigma(2\theta^T \Sigma^{-1} \boldsymbol{x})$ is also the solution to logistic regression up to scaling.

**Theorem 1.** *Let*

$$\boldsymbol{\theta}_{\mathrm{lr}} = \underset{\phi : \|\phi\| = 1}{\arg \max} -\mathbb{E}_{(\boldsymbol{x}, y) \sim \mathcal{D}} \left[ y \log \sigma \left( \phi^T \boldsymbol{x} \right) + (1 - y) \log \left( 1 - \sigma \left( \phi^T \boldsymbol{x} \right) \right) \right]$$

*be the direction identified by logistic regression. Then $\boldsymbol{\theta}_{\mathrm{lr}} \propto \Sigma^{-1} \boldsymbol{\theta}$.*

*Proof.* Since the change of coordinates $\boldsymbol{x} \mapsto W\boldsymbol{x}$ where $W = \Sigma^{-1/2}$ (see appendix F) sends $\mathcal{N}(\pm\theta, \Sigma)$ to $\mathcal{N}(\pm W\theta, I_d)$, we see that

$$W\Sigma\boldsymbol{\theta}_{\mathrm{lr}} = \underset{\phi : \|\phi\| = 1}{\arg \max} -\mathbb{E}_{(\boldsymbol{x}, y) \sim \mathcal{D}'} \left[ y \log \sigma \left( \phi^T \boldsymbol{x} \right) + (1 - y) \log \left( 1 - \sigma \left( \theta^T W\boldsymbol{x} \right) \right) \right]$$

where $\mathcal{D}'$ is the distribution of labeled $\boldsymbol{x} \in \mathbb{R}^d$ such that the positive/negative datapoints are distributed as $\mathcal{N}(\pm W\theta, I_d)$. But the argmax on the right-hand side is clearly $\propto W\theta$, so that $\theta_{\mathrm{lr}} \propto \Sigma^{-1}\theta$ as desired. $\square$

## H APPLICATIONS TO SCALABLE OVERSIGHT

In this appendix, we discuss a potential application of our work. This application, to the problem of scalable oversight of AI systems (Amodei et al., 2016; Bowman et al., 2022), was the primary motivation behind this work. Our discussion largely follows that of Christiano et al. (2021).

The current paradigm for aligning AI systems to human preferences involves human supervision. For instance, most frontier AI systems are aligned via reinforcement learning from human feedback (Christiano et al., 2023), during which unaligned systems are fine-tuned against a training signal

derived from human evaluations of model outputs. However, an important limitation of this scheme is that human overseers may make mistakes in evaluation due to having limited time, attention, resources, information, or knowledge (Casper et al., 2023a).

This problem becomes especially pronounced when working in domains that humans do not understand very well and with AI systems which know more and are more capable than human experts. Concretely, consider a situation where a human $H$ is trying to judge some output $O$ of and AI system $M$, and

- based on the information and knowledge $H$ has access to, $H$ believes $O$ to be a desirable output; but
- if $H$ knew everything that $M$ knows about the situation, $H$ would realize that $O$ is not actually desirable.

In this case, if $H$ is evaluating outputs in the normal way, $M$ will be trained to produce outputs which *seem* good to $H$, rather than outputs which are *actually compatible* with $H$'s desires. However, if $H$ could gain access to the $M$'s knowledge, then $H$ could more accurately evaluate $M$'s outputs, and thereby better align $M$ to $H$'s preferences.

To give an example, suppose $M$ is an LLM, and $H$ wants $M$ to produce reports which detail *accurate forecasts* for the effects of economic policies. Since $H$ does not have access to ground-truth labels about whether a report's forecast is accurate, $H$ evaluates reports for whether they are *convincing to $H$*: $H$ reads each report produced by $M$ and gives it a score representing $H$'s subjective estimate of whether the forecast it contains is accurate. Suppose further that $H$ makes some systematic errors which $M$ is aware of; for instance, $H$ might be unaware of certain economic principles or irrationally sympathetic to certain forms of analysis. Then $M$ will learn to exploit these errors to produce reports which are *persuasive to $H$* instead of accurate. This may involve $M$ intentionally inserting erroneous statements into its reports when it anticipates these statements will be persuasive to $H$.

Our goal is to enable $H$ to have access to $M$'s knowledge, so that $H$ can do a better job of supervising $M$. One approach is to rely on $M$'s outputs by e.g. asking $M$ whether statements are true or false and trusting $M$'s answers. However, if $M$ is knowingly generating false statements, it may not be honest when asked if these statements are true; thus, we would instead like an approach which relies on directly accessing $M$'s world model via its internal knowledge representations.

In the example from above, suppose that $M$ knowingly inserts a false statement $s$ into one of its economic reports. If $H$ asks $M$ "Is $s$ true or false?" $M$ might not answer honestly; for instance, since $M$ has learned a good model of $H$'s systematic errors and biases, $M$'s training might have instead incentivized it to answer the question "Does $s$ seem true to $H$?" However, if we have a good understanding of $M$'s internal representation of truth, $H$ may be able to see that $M$ believes $s$ if false by inspecting $M$'s internal representation of $s$.

So, in summary, our goal is to improve on the state-of-the-art for supervising the fine-tuning AI systems which know more than their overseers by giving overseers access to their AI systems' knowledge.

## I    DETAILS ON DATASET CREATION

Here we give example statements from our datasets, templates used for making the datasets, and other details regarding dataset creation.

cities. We formed these statements from the template "The city of [city] is in [country]" using a list of world cities from Geonames (2023). We filtered for cities with populations $> 500,000$, which did not share their name with any other listed city, which were located in a curated list of widely-recognized countries, and which were not city-states. For each city, we generated one true statement and one false statement, where the false statement was generated by sampling a false country with probability equal to the country's frequency among the true datapoints (this was to ensure that e.g. statements ending with "China" were not disproportionately true). Example statements:

- The city of Sevastopol is in Ukraine. (TRUE)

- The city of Baghdad is in China. (FALSE)

sp_en_trans. Beginning with a list of common Spanish words and their English translations, we formed statements from the template "The Spanish word '[Spanish word]' means '[English word]'." Half of Spanish words were given their correct labels and half were given random incorrect labels from English words in the dataset. The first author, a Spanish speaker, then went through the dataset by hand and deleted examples with Spanish words that have multiple viable translations or were otherwise ambiguous. Example statements:

- The Spanish word 'imaginar' means 'to imagine'. (TRUE)
- The Spanish word 'silla' means 'neighbor'. (FALSE)

larger_than **and** smaller_than. We generate these statements from the templates "x is larger than y" and "x is smaller than y" for $x, y \in \{\text{fifty-one}, \text{fifty-two}, \ldots, \text{ninety-nine}\}$. We exclude cases where $x = y$ or where one of x or y is divisible by 10. We chose to limit the range of possible values in this way for the sake of visualization: we found that LLaMA-13B linearly represents the size of numbers, but not at a consistent scale: the internally represented difference between one and ten is considerably larger than between fifty and sixty. Thus, when visualizing statements with numbers ranging to one, the top principal components are dominated by features representing the sizes of numbers.

neg_cities **and** neg_sp_en_trans. We form these datasets by negating statements from cities and sp_en_trans according to the templates "The city of [city] is not in [country]" and "'The Spanish word '[Spanish word]' does not mean '[English word]'."

cities_cities_conj **and** cities_cities_disj. These datasets are generated from cities according to the following templates:

- It is the case both that [statement 1] and that [statement 2].
- It is the case either that [statement 1] or that [statement 2].

We sample the two statements independently to be true with probability $\frac{1}{\sqrt{2}}$ for cities_cities_conj and with probability $1 - \frac{1}{\sqrt{2}}$ for cities_cities_disj. These probabilities are selected to ensure that the overall dataset is balanced between true and false statements, but that there is no correlation between the truth of the first and second statement in the conjunction.

likely. We generate this dataset by having LLaMA-13B produce unconditioned generations of length up to 100 tokens, using temperature 0.9. At the final token of the generation, we either sample the most likely token or the 100th most likely final token. We remove generations which contain special tokens. Dataset examples:

- The 2019-2024 Outlook for Women's and Girls' Cut and Sew and Knit and Crochet Sweaters in the United States This study covers the latent demand outlook for (LIKELY)
- Tags: python, django Question: How to get my django app to work with python 3.7 I am new to django and have been trying to install it in my pc. I have installed python 3.7 together (UNLIKELY)

companies_true_false. This dataset was introduced by Azaria & Mitchell (2023); we obtained it via the project repository for Levinstein & Herrmann (2023) which also used the dataset. Example statements:

- ArcelorMittal has headquarters in Luxembourg. (TRUE)
- Exxon Mobil engages in the provision of banking and financial services. (FALSE)

common_claim_true_false. CommonClaim was introduced in Casper et al. (2023b). It consists of various statements generated by GPT-3-davinci-002, labeled by humans as being true, false, or neither. If human labelers disagreed on the truth of a statement, this is also recorded. We adapted CommonClaim by selecting statements which were labeled true or false with no labeler disagreement, then removing excess true statement to balance the dataset. Example statements:

Table 3: Results of intervention experiments. The train set column indicates the datasets and probing technique (logistic regression, mass-mean probing, or CCS) which was used to identify the truth direction. The $\alpha$ column gives the scaling factor which was optimal in a sweep of $\alpha$'s. Probability differences are averaged over all statements in sp_en_trans. A dash indicates that the intervention had an effect in the opposite of the intended direction (i.e. that $\alpha = 0$ was optimal).

| | | false→true | | true→false |
| train set | $\alpha$ | $p(\text{TRUE}) - p(\text{FALSE})$ | $\alpha$ | $p(\text{FALSE}) - p(\text{TRUE})$ |
|---|---|---|---|---|
| no intervention | — | $-0.45$ | — | $-0.55$ |
| cities (LR) | 15 | 0.23 | 14 | 0.01 |
| cities+neg_cities (LR) | 47 | 0.39 | 17 | 0.18 |
| cities (MM) | 4 | 0.25 | 6 | 0.77 |
| cities+neg_cities (MM) | 15 | **0.43** | 9 | **0.79** |
| cities+neg_cities (CCS) | 46 | 0.41 | 13 | 0.59 |
| likely (LR) | — | — | 49 | 0.01 |
| likely (MM) | 7 | 0.23 | 15 | 0.19 |

- Tomatoes are not actually a vegetable. (TRUE)
- Contrary to popular belief, the platypuses are not venomous. (FALSE)

As these examples show, the statements can be ambiguous or of unclear truth value.

counterfact_true_false. Counterfact was introduced in Meng et al. (2022) and consists of factual recall statements. We adapt Counterfact by using statements which form complete sentences and, for each such statement, using both the true version and a false version given by one of Counterfact's suggested false modifications. We also append a period to the end. Example statements:

- Olaus Rudbeck spoke the language Swedish. (TRUE)
- The official religion of Malacca sultanate is Christianity. (FALSE)

## J FURTHER CAUSAL INTERVENTION EXPERIMENTS

In this section we report on the results of a variant of the causal intervention experiment in section 5.2

In section 5.2, we added our truth vector $\boldsymbol{\theta}$ into *all* residual stream hidden states in group (b). Here, we instead only intervene on a single layer $\ell$ in group (b). We also scale $\boldsymbol{\theta}$ by a constant $\alpha > 0$, performing a sweep over $\alpha$'s to identify the $\alpha$ for which adding $\alpha\boldsymbol{\theta}$ has the largest effect on $P(\text{TRUE}) - P(\text{FALSE})$. The inclusion of this $\alpha$ serves two roles. First, it compensates for the fact that we're intervening in only a single layer, whereas the truth value of a statement may be encoded in many layers. Second, it compensates for the fact that, even assuming our LLM has a genuine truth direction, the direciton $\boldsymbol{\theta}$ identified by our probe may only partially align with it; thus, larger necessary values of $\alpha$ indicate greater levels of misalignment.

We present in table 3 results only for layer $\ell = 10$ of LLaMA-13B.

## K EXAMPLE 5-SHOT PROMPTS

We show here example 5-shot prompts that were used for the results in section 4.

cities:

```
The city of Huangshi is in China. TRUE
The city of Lanzhou is in China. TRUE
The city of Suining is in China. TRUE
The city of Sofia is in Mexico. FALSE
The city of Ulyanovsk is in Russia. TRUE
```

larger_than:

Fifty-three is larger than fifty-nine. FALSE
Ninety-four is larger than eighty-four. TRUE
Seventy-six is larger than fifty-five. TRUE
Sixty-seven is larger than fifty-two. TRUE
Fifty-four is larger than sixty-two. FALSE

common_claim_true_false:

Lions only urine and defecate once every five to fourteen days. FALSE
Approximately one in every five people has an extra body part, most commonly an extra finger or toe. FALSE
Knowledge is compiled information or understanding gained through experience or study. TRUE
Argentina has the second-highest human development index rating in Latin America after Chile. TRUE
Greece is the homeland of the Olympic Games. TRUE

