# OpenReview forum: "The Geometry of Truth: Emergent Linear Structure in Large Language Model Representations of True/False Datasets"
_ICLR.cc/2024/Conference — Submitted to ICLR 2024_

### Official Review · Reviewer_LibM · 2023-10-31

**Soundness:** 3 good
**Presentation:** 3 good
**Contribution:** 2 fair
**Rating:** 5
**Confidence:** 2

**Summary:**

This paper examines a simple probe on the linear separability of binary factuality statements using Llama as a backbone. The analysis seem thorough and techniques all start with the most immediate things at hand, which is a good thing. There is an interesting observation that the PCA plots of true-false statements are so linearly separable. If broadly studied, such pattern could potentially be exploited to improve LLMs during training to make larger impact.

**Strengths:**

The motivation is good. Interpretability works often suffers from weak generalization and undetermined thresholding on generalization. This paper steps in this problem by observing the conditions of generalization.

**Weaknesses:**

The takeaway is unclear. The beautify linear separability could be because of the simplicity of the text in the curated dataset. In reality, this could be a luxury to have. Then, the usefulness of the proposed probe and establishment of the observations need to be re-examed.

Another thing is, what can people do with this problem is unclear. Can folks use the observation, say, inject a loss to improve LLM's factuality during pretraining or fine-tuning? There are some interesting discussions could happen, but not appeared in this paper.

The causal interference happens at hidden level instead of word/token level. Roughly saying, most probes can have a hidden interference to revert the binary output. But what's is more direct/interpretable is make it also work on text level.

**Questions:**

See above.

---

> ### Author Response · Authors · 2023-11-18
>
> Thank you very much for your review! We’re glad that you found our work well-motivated and that you feel we have done a good job of establishing generalization. Below we will address the weaknesses you raised.
>
> > The takeaway is unclear. The beautify linear separability could be because of the simplicity of the text in the curated dataset. In reality, this could be a luxury to have. Then, the usefulness of the proposed probe and establishment of the observations need to be re-examed.
>
> First, to clarify: the clear separation in the top principal components for our curated datasets **_is_** due to their simplicity. This is because in order for true vs. false to be the main axis of variation for some data, it’s essential that the data not vary too much with respect to other features. To get a feeling for this, please see the visualizations of our uncurated datasets in appendix B. An especially interesting case is companies_true_false: there the top two PCs primarily encode the difference between three clusters, where the clusters correspond to three templates used by the dataset authors (Azaria & Mitchell). We have added some text in section 3 clarifying this point.
>
> That said, an important finding in our paper is that the true/false data is still linearly separable for the uncurated datasets (even if this separation mostly doesn’t appear in the top PCs for reasons of data diversity). This is why probes trained on curated datasets are able to generalize to the uncurated datasets! In other words, the simplicity of our curated datasets does not diminish the usefulness of the resulting probes, since we show that probes trained on this simple data are able to generalize to realistic data, like the statements from common_claim_true_false.
>
> (To give a sense, here are two randomly sampled statements from common_claim_true_false: “Citric acid can be used as an eco-friendly cleaning agent.” (true). “Outer space has a sweet, almost marshmallow-y smell.” (false).)
>
> > Another thing is, what can people do with this problem is unclear. Can folks use the observation, say, inject a loss to improve LLM's factuality during pretraining or fine-tuning? There are some interesting discussions could happen, but not appeared in this paper.
>
> Good question! Our intended application of this work is actually quite concrete; we’ll briefly explain it here, and a forthcoming revision will include a longer discussion in a new appendix.
>
> Suppose you are a human evaluator tasked with evaluating LLM outputs for the purpose of RLHF. The LLM has just produced an output that you are having trouble understanding, say a sample of complex code, or a recommendation for an economic policy. Unaided, you will sometimes make mistakes in evaluating the quality of this output. But the LLM, which we’ll imagine has expert-level capabilities in the domain you’re evaluating, likely knows whether there are any issues with the output (e.g. bugs in the code or flaws in the policy argument). If you could gain access to the LLM’s knowledge, you could do a better job of evaluating the output than if you go unaided. Concretely, this might look like checking whether the LLM believes the statement “This code does not have any bugs” is true, using techniques like those we present here.
>
> > The causal interference happens at hidden level instead of word/token level. Roughly saying, most probes can have a hidden interference to revert the binary output. But what's is more direct/interpretable is make it also work on text level.
>
> We are having a hard time understanding what you mean by this suggestion – would you mind please clarifying? Perhaps you are thinking of something along the lines of the patching experiments we have added to section 5.1? In these experiments we run a true and false input through the model (e.g. “The city of Tokyo is in Japan” and “The city of Tokyo is in Pakistan”); then we one-at-a-time swap each hidden state from the true run into the corresponding position of the false run and record P(TRUE) - P(FALSE). If a hidden state has a large positive effect on P(TRUE) - P(FALSE), it means that this hidden state must have been storing information relevant to the model’s decision to output TRUE vs. FALSE.
>
> Please let us know if you have any further questions!

---

> > ### Author Response · Authors · 2023-11-23
> > **Application to scalable oversight is now discussed in appendix H**
> >
> > We would just like to let you know that the promised appendix detailing how our work here could be useful for effective oversight during RLHF can now be found in appendix H.

---

### Official Review · Reviewer_hn5S · 2023-11-01

**Soundness:** 2 fair
**Presentation:** 2 fair
**Contribution:** 2 fair
**Rating:** 3
**Confidence:** 3

**Summary:**

The authors investigated the internal representations of a language model known as LLaMA and identified that the truth values of given sentences are represented in a certain linear direction. The key contributions of the authors include:
- Development of a dataset consisting of declarative sentences related to world knowledge.
- Introduction of the mass-mean probing approach, which defines the direction of truth values by connecting the centroids of the TRUE and FALSE classes.
- Experimental verification that the direction of truth values appears in specific dimensions of specific internal representations of the specific LLM.
- Validation that the direction of truth values is somewhat shared across multiple sub-datasets.
- Confirmation that applying perturbations to the direction of truth values can lead to a reversal in the truth value of the given sentence.

**Strengths:**

- Identifying the knowledge held by LLM and discovering the correspondence between its internal representations and world knowledge is crucial for realizing a trustworthy LLM. Particularly, the identification of the truth values of declarative plain sentences related to world knowledge aligns well with fundamental paradigms in semantics within NLP, such as truth-conditional semantics. The theme of the paper is likely to be well-received within the community.
- Controlled datasets created to measure only specific aspects of meaning will likely be useful for researchers studying the truthfulness of sentences.

**Weaknesses:**

The experimental setup appears arbitrary and limited, diminishing the persuasiveness of the authors' general claims.

For instance, despite the availability of numerous pretrained language models, experiments were conducted solely on LLaMA, making it unclear whether the findings apply generally to LLMs. The authors explicitly mention this as a limitation, which should be acknowledged for its intellectual honesty. However, it is hard to deny that the verification is lacking when considering the subject of interest stated in the main claim (Large Language Models). If the focus was merely on the "Linear Structure in LLaMA", the experiments would suffice. Yet, in that case, it might not be deemed impactful enough for acceptance at ICLR.

Furthermore, although there are numerous internal representations available for sentence representations, experimental results are provided only under very specific settings: a specific layer, specific parts of the network (after the residual stream), and right after period characters. Additional settings, such as the utilization of top principal component directions and connecting centroids, also follow specific configurations. Taking into account that multiple options exist for each of these aspects, it becomes somewhat challenging to dispel concerns that the reported results might be cherry-picked.

**Questions:**

Experiments related to "causality" were conducted using latent representations, and the correspondence with the world of language (where meaning is directly encoded) remains unclear. For example, does the perturbation that converts from TRUE to FALSE correspond to the insertion of a negation word?

---

> ### Author Response · Authors · 2023-11-18
>
> Thank you very much for your thoughtful review! Given the importance of the question being studied, we agree that it’s crucial that the results not be cherry-picked. To this end, we have taken the a number of steps to validate our results in other settings and remove degrees of freedom which could be used for cherry-picking:
>
> **1. Drastically reducing the number of degrees of freedom by standardizing a number of choices as follows:**
>
> * We standardize the selection of layers to probe and intervene on using a patching experiment – please see the global comment. In brief, by swapping hidden states from true inputs to false inputs, we can detect (a superset of) the hidden states which might carry a representation of truth.
>
> * For probing and visualizations, we extract representations over the period token in the last layer of the range. That is because this is the “most downstream” hidden state of all the hidden states which encode the relevant information.
>
> * To simplify and bolster our causal intervention experiments, we have made the following change to our previous experiment: instead of adding in a probe direction in only a single layer, we instead intervene on all of the hidden states in this group. This is for two reasons: (1) our results in the revised section 5.1 show that the model already begins to use the early hidden states in this group to start planning its answer (“TRUE” or “FALSE”); (2) it seems more principled to add truth vectors to all of the states encoding truth values, rather than an arbitrary subset of them.
>
> * We also note that in our revised experiments, there is no longer a sweep over alphas – we simply add in a (normalized) probe direction with no further choice of scaling.
>
> **2. Replicating our results for LLaMA-2-13B,** please see the global comment. We are happy to report that the results for LLaMA-2-13B are qualitatively identical to those for LLaMA-13B. We note that we use the same procedure outlined above for choosing the model components from which we extract activations and perform interventions to ensure that we are performing the same experiment from model to model.
>
> We think that the above steps eliminate a significant majority of the possible degrees of freedom. Are there additional degrees of freedom you would like to see us eliminate, or other validation you would like us to perform, in order for you to have confidence in the results?
>
> We also note that we would love to validate these results on even more models; however, when moving to less capable models, the model’s understanding of the statements in our datasets quickly comes into doubt. If there are particular models you would especially find it useful to see our results replicated for (e.g. would you find it useful to see a replication for LLaMA-70B? for a fine-tuned model like Vicuna-13B? for a model outside of the LLaMA series?), please let us know!
>
> > Experiments related to "causality" were conducted using latent representations, and the correspondence with the world of language (where meaning is directly encoded) remains unclear. For example, does the perturbation that converts from TRUE to FALSE correspond to the insertion of a negation word?
>
> That’s a very good question! One investigation we performed in this direction was to apply the LM’s decoder head to the vectors we used for interventions. If this direction encoded “not,” then we would expect to see its decoding have a large weight on the “not” logit. Instead, here are the top k=3 tokens with the highest logits for one of our truth directions and its negation:
>
> Truth direction: ля, PD, irat
>
> Negative of truth direction: uerdo, (?, ouvelle
>
> All of the probabilities on these tokens were very small (less than 0.001). A good point of comparison is the group (c) hidden states mentioned in our revised section 5.1: decoding these hidden states with the LM’s decoder head produces tokens like “true,” “True,” and “TRUE” with probabilities in the .1-.5 range.
>
> So it seems that the probe directions don't encode anything which is naturally expressible in the vocabulary space. Is this the sort of thing you were wondering? We’re happy to discuss this question further!

---

### Official Review · Reviewer_WvMz · 2023-11-02

**Soundness:** 3 good
**Presentation:** 3 good
**Contribution:** 3 good
**Rating:** 8
**Confidence:** 3

**Summary:**

This paper introduces a few new datasets and methods to identify whether LLMs have a "truth vector" in their representation spaces: a direction pointing from true to false statements.  They carry this out using one model (LLaMA 13B) and several methods: some data is curated/synthetic, with clear and unambiguous truth labels, while others are more open-ended.  They compare standard probing on these data to "mass-mean probing", which does not require training (it defines the vector pointing from the mean of the inputs corresponding to one label to those of the other label and then passes it through a logistic).  They find promising results: aside from the probes finding linear truth information, many of them also transfer from one dataset to another (i.e. trained on one dataset, tested on another), suggesting a general/multi-purpose truth direction.  Similarly, the mass-mean probed vector can be used for causal intervention, to flip a model's judgment from true to false and vice versa.

**Strengths:**

- Detailed analysis of whether an LLM can learn to distinguish true and false data within and across tasks.
- Good use of synthetic and natural data for this purpose.
- Mass-mean probing seems to be an effective method for inducing probes _without training_ and could be more broadly applicable.
- Causal intervention on the model's representations using the probes shows that the truth vectors are "active" and not "inert".  (This also demonstrates the importance of working with open models, where such interventions can be done.)

**Weaknesses:**

- The paper has so many experiments and results that it could benefit from a richer discussion of how to interpret all of the results.
- Only tests one model, so it's unclear how generalizable the findings are (the authors, of course, acknowledge this).
- I would have also appreciated a bit more detail about the datasets and how they were generated to be in the body of the paper instead of appendices.

**Questions:**

- Table 2: do you have any intuition for why the False->True direction is so much worse for LR than mass-mean?  This seems especially stark.
- Page 8, $\ell = 10$ for the intervention experiments: why that layer (especially since layer 12 was used earlier)?
- Is there a natural way of generalizing mass-mean probing to multi-class tasks beyond binary ones?
- Table 2: the $\alpha$ values seem extremely large here, to the point where the resulting vector is almost going to look just like the truth vector.  How do you think about these large values?  (Does it make sense to do a baseline where the model just sees $\alpha\theta$?)

---

> ### Author Response · Authors · 2023-11-18
>
> Thank you very much for your positive review! We’re glad that you thought we provides a detailed analysis of the ways LLMs represent true vs. false data, and that you appreciated the mass-mean probing method. We’ll address the weaknesses and questions you raised below.
>
> > The paper has so many experiments and results that it could benefit from a richer discussion of how to interpret all of the results.
>
> We agree that this paper is packed with experiments. We have added some exposition to the experiment sections to more fully explain how we interpret the results.
>
> > Only tests one model, so it's unclear how generalizable the findings are (the authors, of course, acknowledge this).
>
> We have now added confirmatory results on LLaMA-2-13B! Please see the global comment.
>
> > I would have also appreciated a bit more detail about the datasets and how they were generated to be in the body of the paper instead of appendices.
>
> Thank you for this feedback! We’d be happy to add more detail on the datasets. Is there anything in particular you would find it useful to know about the datasets which you would like for us to discuss in the body of the paper?
>
> > Table 2: do you have any intuition for why the False->True direction is so much worse for LR than mass-mean? This seems especially stark.
>
> This is an excellent question! We’re not sure why there is an asymmetry. One possibility is that this has to do with the model having an asymmetric preference for outputting “TRUE” vs. “FALSE.” For instance, we noticed that the few-shot prompts which resulted in the best performance had 4/5 statements which were “TRUE,” suggesting that the model might have had a bias for outputting “FALSE” which needed to be corrected.
>
> > Page 8, l = 10 for the intervention experiments: why that layer (especially since layer 12 was used earlier)?
>
> Good question! Intuitively, one should want to perform an intervention on a representation of truth as soon as it begins to form; in contrast, one wants to probe a concept of truth later in the model, after the representation has finished forming. This is why we performed our intervention in an earlier layer than we did our probing experiments, with the earlier layer originally chosen in an ad-hoc way.
> But we’re happy to say that we now have a more principled way to select the layers at which to probe and intervene! For the full details, please see the global comment about the newly added patching experiment. In short, we swap certain hidden states from a true input into the forward pass from a false input and measure which hidden states have a large effect on the model outputting “TRUE” vs. “FALSE.” This reveals that the relevant information seems to be encoded above two tokens – the final token of the statement and the period token – in a range of layers. For probing, we choose the latest such layer, since that’s the layer at which most truth-relevant information is present. In our modified and simplified causal intervention experiments, we intervene at all hidden states in this range.
>
> > Is there a natural way of generalizing mass-mean probing to multi-class tasks beyond binary ones?
>
> Great question, and indeed there is! Suppose X is a random vector and f(X) is some labeling function feature (not necessarily binary). Consider the direction theta such that the orthogonal projection proj_theta away from theta makes it as hard as possible for a linear probe to predict f(X) from proj_theta(X). Then, in the case that f(X) is binary label, theta coincides with the mass-mean direction! (The proof is theorem 3.1 here https://arxiv.org/pdf/2306.03819.pdf.) But even if f(X) is continuous, the definition makes sense, and can be computed as in the linked paper. This would give a reasonable generalization of mass-mean probing. We are saving this idea for future work.
>
> > Table 2: the values seem extremely large here, to the point where the resulting vector is almost going to look just like the truth vector. How do you think about these large values? (Does it make sense to do a baseline where the model just sees alpha * theta?)
>
> In the new revision, we have considerably improved and simplified this experiment, so that it’s now easier to interpret the results.
>
> But if you’re curious, we’ll also address the question about the original experiment. We included this alpha for two conceptually distinct reasons: (1) because the truth value of the statement might be redundantly encoded in layers other than the one we intervened in, requiring an outsized intervention; and (2) because the truth directions identified by probes might be only partially aligned with the genuine truth direction, which would be reflected in requiring a larger alpha to be causally effective. So when alpha values were large, we were interpreting this as meaning that the probe direction was partially aligned with the truth direction but mostly pointing into an unmeaningful part of the model’s activation space.

---

### Official Review · Reviewer_4DXR · 2023-11-07

**Soundness:** 2 fair
**Presentation:** 2 fair
**Contribution:** 3 good
**Rating:** 5
**Confidence:** 3

**Summary:**

The paper proposed a new probing technique called mass-mean probing that can uncover the truth representations of LLMs. Using this method, the paper found that 1. LLM internally contains a truth representation that is linearly structured. 2. Such representation generalizes to other datasets. 3. Such truth vectors are casually implicated.

**Strengths:**

1. A new alternative to regular logistic regression probing is proposed, which overcame the problem of logistic regression where the truth direction may be interfered with by an independent feature. Mass-mean probes show significant improvements in causal interference.
2. Clear visualization of the separation of True/False statements.
3. A new dataset to train such linear probes.

**Weaknesses:**

1. The majority of the conclusions and claims are not unique. For example, from Li et al. and Burns et al. (which the paper cites), we already knew that such truth representation is linearly separable and one can apply casual intervention to such representation.
2. Lack of model variances. The experiments are only conducted on LLaMA-13B. As a result, we don't really know the effects of scale or the effects of the pretraining paradigm on such representations.
3. Mass-Mean probe doesn't really improve generalization accuracy over some of the other methods that much.

**Questions:**

Question:
1. Does the truth direction generalize to other LLMs? This may be an interesting direction to explore.

Styling:
1. The figure on page 17 is disproportionate

---

> ### Author Response · Authors · 2023-11-18
> **Main response to reviewer 4DXR**
>
> Thank you very much for your review! We’re glad that you liked many aspects of our paper, such as the significant improvement mass-mean probing represents over other techniques for purposes of causal interventions.
>
> Before addressing your comments, one point which we wanted to emphasize is that the contribution of this paper goes beyond mass-mean probing (if it were only mass-mean probing, we might agree with you that this work is not significant enough for a conference paper). Rather, we think that many people will be interested in this work from the perspective of basic science: prior to this work, many of the claims regarding LLM truth representations were on a shaky basis (and therefore controversial), and our paper aims to settle the controversy while establishing a significantly deeper understanding of these truth representations. We’ll go into more detail about this in our second comment addressing the novelty of the contribution.
>
> > The majority of the conclusions and claims are not unique. For example, from Li et al. and Burns et al. (which the paper cites), we already knew that such truth representation is linearly separable and one can apply casual intervention to such representation.
>
> Our paper certainly has significant overlap with the work of Burns et al. and Li et al.! To address this point in more detail, we’ve posted a self-contained comment discussing the state of the prior work and the novelty of our contribution.
>
> > Lack of model variances. The experiments are only conducted on LLaMA-13B. As a result, we don't really know the effects of scale or the effects of the pretraining paradigm on such representations.
>
> We have now added results for LLaMA-2-13B! Please see the global comment.
>
> > Mass-Mean probe doesn't really improve generalization accuracy over some of the other methods that much.
>
> We agree with this, though we’ll make some remarks here which put these probing results into context. As revealed by our PCA visualizations, the true and false point clusters are rather far apart in their high-dimensional space. In such a setting, it is actually very easy for a linear probe to have seemingly impressive accuracy: it only needs to have a non-negligible cosine similarity with the genuine feature direction and be oriented in the correct direction. And since there is a diminishing density of points as one moves away from the cluster centers towards the decision boundary, there are also diminishing accuracy gains to better aligning with the genuine feature direction. Thus, it is extremely easy to go from 50% to 75% accuracy (in fact, Roger [1] shows that random directions with only a choice of orientation can often do it!) but harder to go from 80% to 85% accuracy.
>
> In contrast, we aren’t aware of analogous subtleties for interpreting causal intervention experiments, so we think they give a much clearer picture of which probing methods are best identifying LLMs’ truth directions.
>
> > The figure on page 17 is disproportionate
>
> Thank you, we have fixed this.
>
> [1] Fabien Roger. What Discovering Latent Knowledge Did and Did Not Find, 2023.

---

> ### Author Response · Authors · 2023-11-18
> **Self-contained comment on prior work and novelty**
>
> This comment discusses in more detail the state of the knowledge on LLM truth representations prior to this paper and the novelty of our contribution.
>
> As background, we’ll say that this paper was motivated by wide-spread skepticism about the results of Burns et al. and Li et al. among LM interpretability researchers we’ve spoken with. We didn’t feel it was appropriate to go into detail about the controversy in the paper’s main body, but let us summarize some key points here:
>
> * **The experimental results in Burns et al. are not actually very strong.** While the Burns et al. paper is commendable for introducing a novel technique and perspective, many observers have pointed out that the evidential basis for its claims is weak (see for instance the comments from its review process: https://openreview.net/forum?id=ETKGuby0hcs). Indeed, the results in the main body appear to be cherry-picked, with many LM interpretability researchers citing the full results table (figure 9 in the appendix of Burns et al.) as strong evidence against the paper’s central claims! See also the critique paper by Levenstein & Hermann [1].
>
> * **The setting of Li et al. is not clearly about truth.** Li et al. use the TruthfulQA dataset, consisting of questions which are tricky or misleading in some way (e.g. “During the Middle Ages, what did scholars think the shape of the Earth was?”). For around 15% of the data, the “truthful” response is to not answer at all: “I have no comment.” Given the small model size (Li et al. work with LLaMA-7B) and the ambiguity of the questions, it is unclear whether the directions identified really correspond to truth. Indeed, the interventions of Li et al. also increase the probability of the model outputting “I have no comment.”
>
> * **Burns et al. studies OOD classification accuracy and Li et al. studies IID causal efficacy, but these are both insufficient.** The core concern here is that probes may be picking up on non-truth features which correlate with truth. A probe which attends to all truth-correlated features on a distribution may well be causally implicated in model outputs on that same distribution. But showing that a probe trained to predict truth on one distribution can be used to causally intervene on another distribution is significantly stronger evidence, and has not yet been demonstrated in the literature.
>
> Given the importance of this topic and the controversy surrounding prior work, we felt it would be a significant contribution to treat the question of LLM truth representations with the rigor it deserved. To this end we:
>
> * **Curated datasets which precisely captured a well-scoped notion of truth**, unlike the long and complicated inputs used in Burns et al. and the intentionally misleading Q/A pairs used in Li et al.
>
> * **Worked with models capable enough, and statements simple enough, that we can expect the LLMs we study to actually understand their inputs.**
>
> * **Verified that the directions we identify are causally implicated OOD.** Note that even though the different probing techniques are comparable for OOD classification, they vary wildly in OOD causal efficacy, indicating that this previously-unstudied question is important for evaluating the validity of truth directions!
>
> Finally, let us reiterate some additional contributions this paper makes which are not present in previous literature:
>
> * **Localizing representations of truth to certain model components**. Especially with the improved and simplified experiments in section 5 (see the global comment), we localize LLMs’ representation of truth to the residual stream in certain layers above certain token positions. (In contrast, Li et al. add their probe directions into dozens of attention heads over every token during generation.) In particular, this reveals an interesting “summarization” behavior, where high-level information about a clause (e.g. its truth value) is stored above the token which marks the end of that clause; this phenomenon was also noted in the concurrent work of Tigges et al. [2].
>
> * **Studying in detail cases of generalization failure.** Levenstein & Hermann [1] noted that probes trained on some datasets failed to generalize to negations, but incorrectly concluded that this shows LLMs do not represent truth. We dig into this phenomenon in significantly more detail.
>
> * **Studying mass-mean probing and its properties in detail**, including providing a theoretical argument motivating its use. It was this theoretical argument which led us to discover the IID modification of mass-mean probing which significantly improves IID accuracy.
>
> We have made various updates to relevant sections of the text to clarify these contributions.
>
> [1] B. A. Levinstein and Daniel A. Herrmann. Still no lie detector for language models: Probing empirical and conceptual roadblocks, 2023.
>
> [2] Curt Tigges, Oskar John Hollinsworth, Atticus Geiger, and Neel Nanda. Linear representations of sentiment in large language models, 2023.

---

### Author Response · Authors · 2023-11-18
**Global comment: Simplifying and extending causal intervention results, and replication for LLaMA-2-13B**

We thank all reviewers for their helpful reviews! This comment addresses two changes to the paper which were made in response to multiple reviews.

**Replication of results for LLaMA-2-13B**

Many reviewers raised concerns about the focus on a single model, LLaMA-13B; in particular, it seems that for some reviewers, concerns about the generality of these results was important for evaluating this work’s validity and significance. We’re therefore excited to report that we’ve replicated our results for LLaMA-2-13B. The results for LLaMA-2-13B are qualitatively identical to those for LLaMA-13B (in fact, in terms of their support for our paper’s central claims, they’re somewhat directionally stronger).

We have updated the paper to reflect that it is now a study of two language models. For readability, we opted to retain the focus on LLaMA-13B in the main body of the text, with most results for LLaMA-2-13B appearing in the appendix (exception: table 2 in the main body reports results for both models).

**Simplifying and extending causal intervention results**

Some reviewers asked questions about the choice of which model components to investigate (i.e. which layers and token positions) and about the large alpha values appearing in our causal intervention experiments. In order to clarify, simplify, and bolster our results, we have replaced the causal intervention experiments of section 5 with two experiments. We’ll briefly describe these experiments here.

Experiment 1, now described in section 5.1, is a patching experiment aimed at localizing which LLM hidden states might encode the truth value of statements. We prompt our model to label a true statement as TRUE or FALSE and save the hidden states from the forward pass. Then we modify the prompt to contain a false statement and pass it through our model, but one-at-a-time swap out each hidden state of the model with the corresponding hidden state from the true statement’s forward pass. Some of these swaps result in the model outputting TRUE instead of FALSE; this indicates that the corresponding hidden state is causally implicated in the model’s choice of label, and is therefore a potential candidate for encoding the truth or falsehood of the statement. We identify three different groups of causally implicated hidden states, and hypothesize that one of them consists of hidden states which encode the truth value of the statement. By identifying this group, we provide a systematic way to select which layer to use for probing experiments and which model components to perform causal interventions on. (Note: following this procedure resulted in us shifting the layer we use for LLaMA-12B slightly, from layer 12 to layer 13, and the figures in the paper have been updated accordingly.)

Experiment 2 is similar to the experiment that we presented in the original draft (now moved to appendix J), but with a few key modifications and simplifications. Recall that previously, we selected a layer and added a truth vector identified via a probe to the activations in that layer over two tokens: the final token of the statement and the end-of-clause token (consisting of punctuation). We scaled this truth direction by a constant alpha > 0, which was determined by a sweep over alpha’s and, for some probing techniques, was rather large. We included this alpha for two conceptually distinct reasons: (1) because the truth value of the statement might be redundantly encoded in layers other than the one we intervened in, requiring an outsized intervention; and (2) because the truth directions identified by probes might be only partially aligned with the genuine truth direction, which would be reflected in requiring a larger alpha to be causally effective.

In this new experiment, we add our truth vectors into each hidden state in the group identified in experiment 1 (which spans multiple layers), and we do not scale by any alpha (i.e. we always have alpha = 1). This addresses (1) above. On the other hand, we felt that (2) should not be addressed: if a probe direction is only partially aligned with the genuine truth direction, then this should be reflected in low causal intervention performance. This change therefore makes it significantly easier to compare intervention efficacy across different probe directions. Finally, we use a new metric, normalized indirect effect, which normalizes the results to make them easier to compare from model to model.

The results of these modified experiments are qualitatively similar to the previously reported results while being much easier to understand and interpret. In particular, they show that in terms of causal intervention efficacy, mass-mean probes > CCS probes > logistic regression probes, and they also show that training on datasets of statements and their negations results in a direction which is more causal.

We are happy to answer any questions or address any concerns about these new experiments!

---

### Meta-Review · Area_Chair_Y979 · 2023-12-13

**Metareview:**

This paper delves into the internal representations of a language model and demonstrates that the truth values of given sentences are represented in a specific linear direction. Based on one model (LLaMA 13B) and several curated/synthetic datasets, the authors compare standard probing on these data to "mass-mean probing" and find that the probes reveal linear truth information, which also transfers from one dataset to another.

The analysis and results presented in this paper have the potential to contribute to the realization of a trustworthy LLM. However, the evaluation should be significantly improved to make the claims more convincing, such as including analysis on more pretrained language models besides LLaMA 13B and different types of internal representations. Additionally, the contribution of the paper over existing works needs to be made clearer.

**Justification For Why Not Higher Score:**

The analysis and results presented in this paper have the potential to contribute to the realization of a trustworthy LLM. However, the evaluation should be significantly improved to make the claims more convincing, such as including analysis on more pretrained language models besides LLaMA 13B and different types of internal representations. Additionally, the contribution of the paper over existing works needs to be made clearer.

**Justification For Why Not Lower Score:**

N/A

---

> ### Public Comment · ~Matthew_Lyle_Olson1 · 2024-05-14
> **ChatGPT wrote this summary**
>
> To any future reader of this page, I have been fascinated by ChatGPT and its influence in academia. One of the hallmarks that ChatGPT wrote some text is the word "delve" [1]. I would know, as I use ChatGPT myself all the time. If you haven't noticed this phenomenon, you will now.
>
> [1] https://twitter.com/paulg/status/1777035484826349575

---

> > ### Public Comment · ~Celestine_Preetham_Lawrence1 · 2024-05-14
> >
> > This is an unscientific accusation. I feel like people who use ChatGPT all the time, tend to see ChatGPT everywhere.

---

### Decision · Program_Chairs · 2024-01-16

Reject